# Preparation of Surgical Thread from a Bioplastic Based on Nopal Mucilage

**DOI:** 10.3390/polym15092112

**Published:** 2023-04-28

**Authors:** Evelyn Herrera-Ibarra, Mercedes Salazar-Hernández, Alfonso Talavera-López, O. J. Solis-Marcial, Rosa Hernandez-Soto, Jose P. Ruelas-Leyva, José A. Hernández

**Affiliations:** 1UPIIG, del Instituto Politécnico Nacional, Guanajuato 36275, Mexico; eherrerai1601@alumno.ipn.mx (E.H.-I.); rohernandezs@ipn.mx (R.H.-S.); 2Departamento de Ingeniería en Minas, Metalurgia y Geología, División de Ingenierías, Universidad de Guanajuato, Guanajuato 36025, Mexico; merce@ugto.mx; 3Unidad Académica de Ciencias Químicas, Campus UAZ siglo XXI, Universidad Autónoma de Zacatecas, Zacatecas 98160, Mexico; talavera@uaz.edu.mx; 4UPIIZ, del Instituto Politécnico Nacional, Zacatecas 98160, Mexico; ojsolis@ipn.mx; 5Facultad de Ciencias Químico-Biológicas, Universidad Autónoma de Sinaloa, Ciudad Universitaria, Culiacán 80030, Mexico

**Keywords:** biopolymer, degradation, mucilage, bioplastic, surgical thread, solvents

## Abstract

Currently, natural materials represent a sustainable option for the manufacture of biopolymers with numerous industrial applications and characteristics comparable with synthetic materials. Nopal mucilage (NM) is an excellent natural resource for the synthesis of bioplastics (BPs). In the present research, the fabrication of biopolymers by using NM is addressed. Changes in the plasticizer (sorbitol and cellulose) concentration, in addition to the implementation of two sources of starch (corn starch (CS) and potato starch (PS)) to obtain the surgical thread, were analyzed. The NM extracted was close to 14% with ethanol. During the characterization of the extract, properties such as moisture, humidity, viscosity, and functional groups, among others, were determined. In the CS and PS analysis, different structures of the polymeric chains were observed. BP degradation with different solvents was performed. Additionally, the addition of sorbitol and cellulose for the BP mixtures presenting the highest resistance to solvent degradation and less solubility to water was conducted. The obtained thread had a uniform diameter, good elasticity, and low capillarity compared to other prototypes reported in the literature.

## 1. Introduction

The environmental damage derived from anthropogenic activity has accelerated the change in the planet’s climate, and the use of fossil hydrocarbons contributes significantly to this process since they produce greenhouse gases [1,2]. The manufacture of synthetic materials (mainly polymers) from oil refining represents a large part of the products used in homes, industries, shops, and medicines, among other applications [2,3,4]. These plastics cause serious damage to the environment because they are mainly non-reusable products, generating large amounts of solid waste which require decades to degrade [1,2]. Due to this situation, in recent years there has been an increased interest in polymers obtained from renewable sources with features similar to synthetic plastics, giving rise to the development of BPs [3,5]. These BPs are biocompatible, biodegradable, non-toxic, and recycle organic matter from agricultural waste, animals, microorganisms, and plants. Therefore, they are considered cheap, sustainable, and widely available materials [1,2,5]. These advantages allow their use in different industries such as food, medicine, and agriculture, among others [2,3,5]. In the medical industry, BPs are widely used to pack and control the supply of medicine, fixation equipment, adhesion, and biomedical polymers used for suture wounds, cuts, tissue engineering, etc. [3,5,6]. Biomedical polymers are products employed to reproduce the function of living tissues in biological systems in a safe, mechanically functional, and physiologically acceptable way [7]. They are implanted in the body (temporarily or permanently), and they try to restore the existing defect by activating tissue regeneration, decreasing the probability of infection, and minimizing patient discomfort [7,8]. In the case of sutures, many factors must be considered, such as the required tension and the biological reaction to the material that occurs in sutures based on synthetic polymers. Furthermore, natural polymers such as starch and cellulose fulfill these characteristics to achieve a natural thread useful in suture wounds [5,9]. For manufacturing surgical threads, the source of the biopolymer and plasticizers conferring the appropriate properties for a particular application has to be analyzed [10,11,12,13]. Glycerol is a plasticizer in the manufacture of starch films preventing cracking during handling and storage; moreover, it has a good sorption capacity due to water permeability. Therefore, adding glycerol as a plasticizer is essential to improve the toughness of the films [12]. On the other hand, sorbitol is a polyalcohol employed as a plasticizer because it confers better elasticity and resistance to films compared to glycerol [13]. Cellulose is a polysaccharide composed of D-glucose units and is the main component of various plants such as cotton and flax. Cellulose derivatives have several applications due to their mechanical properties and biocompatibility; in addition, cellulose is considered a non-harmful component of the environment. These features are why cellulose fibers have stood out in tissue reconstruction and regenerative medicine [11].

For the manufacture of biopolymers, fibrous polysaccharide materials are preferred. There has been an increasing interest in the nopal, a fruit tree in the arid and semi-arid zones of Mexico, with physiological and morphological characteristics that allow its adaptation to extreme temperature and water scarcity [14,15,16,17]. Nopal leaves excrete mucilage, a highly branched fibrous polysaccharide, whose molecular weight is around 13 × 10^6^ g/mol [18,19,20]. It is a complex carbohydrate since it contains approximately 35–40% arabinose, 20–25% galactose and xylose, and 7–8% rhamnose and galacturonic acid [21,22,23,24,25]. The proportion of these monomers in their composition changes depending on factors such as variety, age, environmental conditions, and structure (fruit, shell, or cladode) used for extraction [22,23]. Nonetheless, mucilage is considered a residue commonly discarded by consumers [8,14,18].

In this work, thread for medical applications was obtained by the mixture of NM, potato starch, and glycerol as a base to form molecular networks. Modifications in the proportion of the components in this mixture led to changes in viscosity, elasticity, hardness, resistance, texture, tension, water retention, and degradation in different solvents of the thread. Additionally, its physical properties (diameter, malleability, and color) changed. Moreover, the effect of adding sorbitol and cellulose as a plasticizer on the properties of the thread was studied.

## 2. Materials and Methods

### 2.1. Reagents and Nopal Harvest

All chemicals used were analytical grade. Two different potato starches (PSs) were employed, one from Sigma Aldrich (St. Louis, MO, USA) (CAS: 9005-25-8) and the other one from Baker Analyzed (CAS: 9005-84-9). Sorbitol (C_6_H_14_O_6_) and cellulose were purchased from Merc (100%, CAS: 50-70-4) and J.T. Baker (DEAE, CAS: 9004-34-6), respectively. The water used for the preparation of all the solutions was deionized. The cladodes (leaves) of nopal (*Opuntia ficus indica*) were collected from a producing area in Silao city, Guanajuato, Mexico, at an altitude of 1780 m above sea level with two types of a sub-humid temperate climate with summer rains and semi-dry. Through the PlantNet identifica^®^, PlantSnap^®^, and PictureThis^®^ applications, which are databases of images and plant data, it was identified whether the nopal collected belonged to the species *Opuntia ficus indica*.

### 2.2. Nopal Mucilage (NM) Extraction

For the NM extraction, the nopal should be two years old because the tissue contains a considerable amount of mucilage [17,18,19,24]. The collected cladodes were brushed to remove spines, washed, and disinfected by immersing in 10% (*v*/*v*) sodium hypochlorite (NaClO) solution for 10 min. The cuticle and epidermis were subsequently removed [18,19]. The remaining material was cut into cubes, submerged in deionized water with a ratio of 1:7 (*wt*/*v*) for 48 h, and was stored in a refrigerator (Thermo Scientific (Waltham, MA, USA) FRGL 1204V) at a temperature of 4 °C. Subsequently, the mixture was crushed in an industrial blender (Tapisa 12 L) and then sieved through filter no. 16 (1.18 mm stainless steel mesh) to remove the rest of the plant tissue. Small plant debris was removed from the viscous fluid with a vacuum pump (THOMAS QR-0080), and then this fluid was heated in a water bath (Julabo Ecotemp TW20) at 80 °C for 1 h. Next, the viscous liquid was allowed to cool at room temperature, followed by the addition of ethyl alcohol (J.T. Baker, ≥96%) in a 1:2 (*v*/*v*) ratio. The resulting solution was heated at 80 °C for 1 h. Finally, the mixture was centrifuged (HERMLE Z383K) for 20 min at 2240 rpm to extract the sediment corresponding to the mucilage [18,19,25]. The efficiency of the extracted NM was calculated with an equation based on the recovered weight of mucilage as follows:(1)$$R_{\mathrm{NM}}=\frac{W_{\mathrm{MO}}}{W_{P}}\times 100$$
where R_NM_ is NM efficiency (%), W_MO_ is the mucilage weight (g), and W_P_ is the initial nopal weight (g).

### 2.3. Starch Extraction from Corn Grains

In the CS extraction, 250 g of corn grains were dried, grounded, and immersed in a 0.1 M NaOH solution for 18 h. Afterward, the mixture was poured and mixed in an industrial blender (Tapisa 12 L) and sieved (0.15 mm) to separate big particles from the solution. These big particles were washed several times with deionized water until the washing water remained crystalline. The solution and the washed particles were centrifuged (HERMLE Z383K) for 30 min at 5000 rpm to discard the supernatant. In the solid phase, there was a gray coloration in the upper zone, indicative of proteins, lipids, and small CS granules. In the lower region, it had a white coloration, allegedly by the presence of starch [23,24]. The obtained starch was submerged in a 0.1 M HCl solution until reaching a pH of 6.5 to neutralize the NaOH in CS. Subsequently, the CS was washed with deionized water to remove the remaining HCl. Then, the solution was centrifuged again under the above-mentioned conditions. Finally, it was filtered with a vacuum pump (THOMAS QR-0080) and dried in a forced convection oven (Shel Lab CE5F) at 45 °C for 24 h, for later storage until needed.

### 2.4. Bioplastics (BPs) Elaboration

The extracted NM was slowly poured into deionized water at 25 °C with vigorous stirring for 24 h and then heated at 35 °C for 12 h. Immediately after, each component (NM, glycerol, CS, and water) was added to the mixture (350 mL) while maintaining the glycerol (0.5 mL) and CS (1 g) constant and modifying the proportions of water, as indicated in Table 1. In each experiment, the solution was heated to 90 °C for 30 min and homogenized for 10 min. After this time, the mixtures were cooled to 25 °C, and the pH was adjusted (between 4.5 and 7.5) with 0.1 M citric acid (C_6_H_8_O_7_) [26]. The BP obtained was poured into Petri dishes and placed in a forced convection oven (Shel Lab CE5F) at 40 ± 2 °C for approximately 24 h to reduce the humidity of the mixtures. Following this, resistance tests of the BP (0.2 g) were carried out using 10 mL of different solvents (water, ethyl alcohol, acetone, HCl (0.1 M) and CH_3_COOH (0.1 M)) by observing the BP solubility in each of the solvents for 12 h.

To perform the experiments of BP solubility in water, first, the material was cut into 4 cm^2^ squares, dried at 100 °C for 24 h, and weighed. Subsequently, the samples were immersed in 100 mL of deionized water for 24 h. Finally, the remaining films were recovered by filtration and dried (100 °C in an oven for 24 h) to determine the non-solubilized dry mass. The solubility percentage was obtained with the following equation [10,26,27]:(2)$$\mathrm{Solubility}\;(\%)=\frac{W_{\mathrm{ds}}-W_{\mathrm{df}}}{W_{\mathrm{ds}}}\times 100$$
where W_ds_ and W_df_ are the initial and final dry weight (g) of the bioplastic, respectively. 

### 2.5. Surgical Thread (ST) Preparation

To the bioplastic with the highest resistance to solvent degradation, sorbitol and cellulose with different concentrations were added. The amounts of plasticizers and starch are shown in Table 2. Finally, this mixture is placed in a syringe (Plastipak 10 mL pharmaceutical grade syringe and hypodermic nursing needle) to produce the thread [5,11,27,28]. 

### 2.6. Characterization of Nopal Mucilage and Bioplastic Additives

Attenuated total reflectance-Fourier transform spectroscopy (ATR-FTIR) analyses on NM, CS, SAPS, and BAPS were conducted in a Thermo Scientific Nicolet iS10 analyzer; 32 scans were obtained with a resolution of 4 cm^−1^ over the wavenumber range of 4000–400 cm^−1^. X-ray diffraction (XRD) patterns were obtained in a diffractometer (Ultima IV Rigaku). Scanning electron microscopy images and X-ray energy dispersion spectroscopy (SEM-EDS) were obtained in a JOEL spectrometer (6510 pus). Adsorption spectra in the UV-Vis region (1100–200 nm) were performed in a UV-Vis-NIR (Near IR) spectrophotometer (Cary 5000 Varian). Textural properties of the different starches were determined using the BET equation (Micromeritics Tristar II plus). The apparent viscosity of NM was measured by a digital Brookfield Viscometer (Generic DH-DJ-8S Model). The mechanical properties of the selected bioplastics were determined using a Texture Analyzer (Shimadzu EZ-LX Short Model). The diameter of the obtained thread was measured with a Digital Vernier (Mitutoyo 150 mm 500-196-20 model).

## 3. Results and Discussion

### 3.1. Nopal Mucilage (NM) Extraction

The cladodes of the nopal were cleaned, disinfected, cut into small cubes, and placed in distilled water (1 g/1 mL ratio), and then the solution was crushed, sifted, and filtered. The solid was suspended with ethyl alcohol (ratio 1:2 *v*/*v*). Figure 1a shows the mixture of NM and ethanol, which was centrifuged to separate the supernatant and then dried (Figure 1b), obtaining a yield of 11.1%.

The NM yield can be affected by different variables, including temperature, washing, solvent, and the age of the nopal [12,19]. The yield of NM in this work was ~62%, a value higher than reports in the literature [12] due to cleaning, washing, cutting the nopal into cubes at a pH of 4.45, and NM separation after the addition of ethanol. These modifications were crucial to increase the yield. It is worth mentioning that it was not required to dry the precipitate obtained because the aqueous solution of NM can be directly used. In the particular case of the use of solvents for the precipitation of NM, yields between 18.8 and 20.9% have been reported using isopropanol (1:4, *v*/*v* ratio) and ethanol (1:3, *v*/*v* ratio) [17]. In comparison, the results of the present study are slightly higher than those reported in the literature [12]. 

The apparent viscosity obtained for the NM in this study was 55.2 mPa∙s at a pH of 5.8. In the literature [29,30], a direct relationship between the pH value and the viscosity of the extract is mentioned, in which increasing the pH increases the viscosity value. A liquid with this behavior is considered a non-Newtonian fluid with characteristics of a pseudoplastic fluid [29,30].

### 3.2. Characterization of the NM and Starch 

The textural properties of the NM extracted had a surface area of 1.76 m^2^/g, with a pore volume and diameter of 0.0081 cm^3^/g and 5.79 nm, respectively, characteristic of a mesoporous structure [31]. These properties are comparable with those reported by Abdullah et al. [32]. Figure 2 shows the ATR-FTIR spectrum of the NM, where the bands observed at 3400 cm^−1^ are attributed to the stretching vibration of the hydroxyl groups (O–H) while at 2925 and 1642 cm^−1^ they are assigned to the symmetric stretching vibrations of the –CH group and polysaccharide molecules, respectively [18,19]. The shoulder at 2870 cm^−1^ is attributed to the symmetric extension vibration of the –CH groups. The band at 1414 cm^−1^ corresponds to the carboxylic acid and peptide bond with the vibrations of deformation C–H and deformation N–H bonds of NM [18,19]. The bands at 1242, 1038, and 890 cm^−1^ are related to the functional groups of polysaccharides and are known as the NM fingerprint [14,15]. The structure (Figure 2), morphology (Figure 3), and surface area of the NM allowed an excellent interaction with the plasticizers and starch, a requirement for the manufacture of the surgical thread [18,19].

The ATR-FTIR characterization of SAPS and CS can be seen in Figure 4. For the PS, the bands appearing at 3575 and 3215 cm^−1^ correspond to the stretching vibrations of the hydroxyl group (–OH); meanwhile, the band at 2935 cm^−1^ is attributed to the stretching vibrations of C–H bonds. The signal at 1660 cm^−1^ is assigned to the extension vibrations of C=O, and the band at 1467 cm^−1^ is due to the extensional vibration of N–H bonds. The bands at 1364 and 1161 cm^−1^ appear due to symmetric C–H bending and asymmetric C–O–C extension, respectively. The signals at 1082 and 982 cm^−1^ are related to the stretching vibrations of C–O–C bonds. The bands at 925, 862, and 769 cm^−1^ are generated by the vibrations of the carbohydrate ring in the C–O–C bond [32,33]. In the case of CS, the bands at 3400 cm^−1^ correspond to the extension vibration of the hydroxyl group (–OH), while the signal at 2935 cm^−1^ is attributed to the extension of C–H bonds. The band at 1655 cm^−1^ is associated with OH groups. The bands at 1460 and 1421 cm^−1^ are consistent with the deformation and symmetric scissoring of CH_2_, respectively. The signals at 1360 and 1168 cm^−1^ are assigned to the symmetric bending of C–H and the asymmetric extension vibration of C–O–C. At the bands at 1075 and 991 cm^−1^, the stretching vibrations of C–O are observed; and at 928, 856, and 762 cm^−1^, the carbohydrate ring vibrations due to the C–O–C bonds appear in the spectra [33].

The micrographs of CS and SAPS are shown in Figure 5. Predominantly, polygonal spherical morphology with a size > 5 μm is observed for the CS. An ellipsoidal oval shape with a size > 20 μm is observed for PS. The shape and size of the starch particles are directly dependent on amylopectin. In the case of CS, there are small starch granules, which allows the structure to have short chains with several ramifications, while for the PS, its chains are long with few ramifications due to its larger granules [23,34]. These characteristics are important because they transcend the physical–chemical, functional, and nutritional properties. For example, there is better digestibility in CS since the size of the granules is related to the viscosity; with smaller granules, there is a lower viscosity. Therefore, the small CS granules are digested at a higher speed when compared to the PS granules [23,33,35].

In Figure 6, the diffractograms of CS and PS are shown. The CS has a type A pattern, and the PS is consistent with a type B pattern. In the diffraction patterns of CS (Figure 6b), there are peaks at 15.2°, 17.4°, 18.0°, 19.9° and 26.7°, characteristic of a crystalline structure of double helixes forming a cell monoclinic unitary type [23,27,33,36,37]. The peaks between 15.0° and 23.0° are related to the amylose content and the degree of crystallinity of the starch granules [23,27,33,36,37]. For the PS diffractogram (Figure 6a), peaks at 5.7°, 15.2°, 17.1°, 19.9°, 22.2°, and 24.1° appear and are characteristic of this tuber indicative of a hexagonal symmetry due to the hydrated double helix of amylopectin [33,36]. The peak at 5.7° suggests a more hydrated and open structure of the polymorphism type [23,27,33,36,37]. In addition, PS has semicrystalline starch granules because a part of the structure consists of amylopectin, and the other part is related to the amorphous crystalline structure of starch (between 25° and 35°) [33,36]. 

PS and CS have similarities and differences in their structure, observed by ATR-FTIR, SEM, and XRD analysis. This characterization allowed the selection of starch to obtain a BP with better features [10,11,12,13]. Based on this analysis, the most suitable choice is CS to elaborate the BP with the designated plasticizers (glycerol, cellulose, and sorbitol).

### 3.3. Manufacture of the Bioplastics (BPs)

For the elaboration of the bioplastic from the NM, CS was added in the proportions shown in Table 1. The resulting bioplastics from these combinations can be seen in Figure 7. In experiments 1, 2, and 4, the bioplastics are almost transparent with a weak adhesion and brittle texture. In test 7, a high presence of humidity caused less malleability and resistance to the tension of the bioplastic. In test 6, the bioplastic presented good hardness, rigidity, resistance, and low malleability, which are not desired features for a bioplastic [2,3,5,28,34]. The properties of the BP discussed so far are not the most appropriate complicating its implementation in the medical industry, specifically in the case of threads for medical use. Furthermore, the BP obtained in experiments 3 (E3) and 5 (E5) had the appropriate malleability, adhesion, and elasticity. These characteristics are essential when the bioplastic is considered for surgical thread use [2,3,5,28].

The bioplastics with unfavorable characteristics for surgical thread manufacture are due to the hydrophilic nature of the starch-containing films, which causes a fragile appearance due to weak intermolecular forces [27,38]. In the elaboration of the polymeric films from NM, glycerol, and starch, as the concentration of the plasticizer rose, the solubility of the biofilm increased. According to the properties required for a surgical thread mentioned in the literature, the bioplastics E3 and E5 had the best characteristics; nonetheless, the thermoplastic properties of the CS and NM film still need to be improved to obtain a more appropriate BP for its application in wound suturing [2,3,5,28,35]. Thus, a plasticizer was added to promote molecular disruption or interruption, triggering film flexibility by reducing the internal hydrogen bonds in the polymer chains while increasing the space between the molecules [27,39]. 

### 3.4. Analysis of Solubility in Solvents

The mechanical resistance of bioplastics is studied by analyzing the polymeric degradation of the material by processes including hydrolytic agents [3,40,41]. Here, the decomposition of the polymer chains of the bioplastics was examined by hydrolysis of the amide bonds. To accomplish this, the resulting bioplastics of experiments 3 and 5 (E3 and E5) were placed in contact with different solvents (Figure 8) at room temperature for 12 h. With this, the mechanical behavior was investigated.

Table 3 contains the results of the solubility of the biofilm. The manufactured bioplastics E3 (B1) and E5 (B2) had high resistance to solubility in ethyl alcohol; on the other hand, the bioplastic was dissolved entirely in acetic acid. With these results, it is possible to mention that the biofilm solubility increases as follows: ethyl alcohol > acetone > hydrochloric acid > water > acetic acid. The depolymerization occurring in the biofilms was due to the hydrolysis reaction (breaking of hydrogen bonds) between the polar groups of the amide ester or urethane bonds of the polymer chains [4,40,41,42,43].

To know which of the two bioplastics is a better choice during the elaboration of surgical thread, measurements of water solubility were conducted. The bioplastic solubility with water is an important property to measure in biodegradable films since some applications of these materials require the preservation of the integrity and resistance to corrosion. The percentage of water solubility for each bioplastic is listed in Table 4. A lower solubility was observed in B2 with water and solvents (except for acetic acid); therefore, it is considered suitable for surgical thread applications.

When comparing our bioplastics solubility results in water, the literature has reported similar values (between 23.20 and 33.37%) with different starch sources. Several studies have reported the water solubility of bioplastics containing glycerol and starch of different origins, such as cassava, wheat, potato, corn, and yam, with solubility values of 24%, 30.2–33.2%, 33.88%, 34.37%, and 32%, respectively. In other studies, the obtained bioplastics used sorbitol and starch extracted from sugar palm, mango, and rice seeds, obtaining a solubility of 31.56 to 37.05%, 29.12 to 37.05%, and 24.70 to 31.60%, correspondingly [4,10,40,41,42,43]. The presence of sorbitol as a plasticizer in the bioplastic increases the solubility and diffusion in water because it changes the mobility chain and makes the polymeric matrix more flexible [4]. Hence, the solubility of the bioplastic containing glycerol, sorbitol, starch, and NM obtained in this study presents a suitable solubility for consideration in thread manufacture for medical applications.

### 3.5. Mechanical Properties of Bioplastics

The analysis of the breaking or bending stress (σi) and Young’s modulus (Ei) performed on B1 and B2 is shown in Table 5. It was observed that B2 has a bending greater than B1. This indicates that there are structural changes in bioplastics due to the formation of agglomerates between the protein and amylase/pectin, which can decrease the plasticization of the biopolymer and reduce the mechanical resistance of the material [44]. It was also observed that the bending of the bioplastic depends directly on the pressure applied to the material. The results of the Young’s modulus of the bioplastics (Table 5) indicate that B2 has a greater capacity to resist the stresses that can deform it, which would prevent it from breaking when handled. Bioplastics obtained with materials of natural origin with glycerol and starch have been reported in the literature, where it is mentioned that this behavior may be due to the adherence of the materials that make up the bioplastic due to its manufacturing process that favors its mechanical resistance [43,45,46]. Based on the results obtained from the bioplastic prototypes obtained, B2 is the best option to produce the thread because it presents greater resistance to traction, tension, and bending, and it must have a lower solubility in water.

### 3.6. Preparation of BP with Starch from Different Sources

Another relevant factor in the manufacture of surgical thread is the origin of the starch, the basis of the bioplastic, given that this source determines the size and shape of the granules, originating modifications of some functional and nutritional properties [19,36]. Figure 9 shows the resulting bioplastics made with starch from different sources of origin. The bioplastic with the best perceptible physical properties was prepared with CS, followed by SAPS. This behavior is due to the shape of the starch granules, which depends directly on the structure of the amylopectin ramification since amylopectin can have long chains with few branches (large granules) and short chains with several branches (small granules) [23,34]. There are reports in the literature stating that corn starch granules have a smaller size than potato starch (ratio 1:4); in addition, the shape of the granules is very different since corn has a polygonal shape and potato starch a spherical oval shape [23]. 

The plasticizer can be related to structural modifications of the resulting network between the starch and NM. The films become denser when the plasticizer concentration is decreased [4,27]. Lower plasticizer concentration hinders the movement of the polymeric chains, increasing the solubility of the biofilm. Glycerol has been used as a plasticizer for starch biofilms due to its compatibility with amylose, improving the mechanical properties of the biofilm by directly interfering with the ordering of amylose. This arrangement reduces intermolecular forces [4,27,41,42,43] due to the hydroxyl groups (–OH) in its structure, allowing glycerol to join through hydrogen bonds with the starch chains, completing their order and avoiding the fragility of the biofilm [42]. During the synthesis of BP, the most commonly used plasticizer is glycerol [4,27,39]; however, with sorbitol, better mechanical resistance to the obtained film is conferred since its moisture content is lower compared with the formed films with glycerol [42]. In the elaboration of BP with sorbitol, an increase in the resistance of the biomaterial is found when mechanical stress is applied, suggesting that sorbitol concentration in the bioplastic should be high to obtain a thread for medical applications [4,27,42].

### 3.7. Preparation of Surgical Thread with Different Plasticizers

The manufacture of thread with the proposed methodology (Table 2) resulted in materials with different characteristics. In the resulting prototypes with SAPS, the materials contained high humidity and viscosity. The excessive water in these materials was hard to remove by drying (at low or high temperatures) or increasing the evaporation time; hence, the materials were discarded. The thread prototypes made from the bioplastic with CS, sorbitol, and different cellulose concentrations (Figure 10a) showed greater tensile strength and consistency (1) when adding the same amount of sorbitol to the mixture (0.58 wt%). In another prototype (2), where the amount of cellulose was increased (0.97 wt%) while maintaining the sorbitol amount in the mixture, the threads had greater flexibility and moisture. For the last prototype (3), the amount of cellulose was 1.93 wt% (twice as much as prototype 2), and these threads were more rigid; nevertheless, the tensile strength diminished. These results suggest that with an increment in cellulose concentration of the mixture, the resulting material is more flexible and humid. However, if the cellulose concentration is higher than 0.97 wt%, resistance to mechanical stress is lost, and the thread becomes hard. 

The prototypes made without sorbitol (Figure 10b) and different cellulose concentrations (1.93 wt%, 2.90 wt%, and 3.89 wt%) with CS presented different characteristics in terms of cellulose content since there is better consistency and resistance to tension in the prototype thread (1). In the other prototype (3), the cellulose concentration varied from 1.93 wt% to 2.9 wt%, and better texture and malleability were observed. Finally, when the cellulose concentration was 3.89 wt%, the thread prototype (2) presented fragility in its structure. Based on the USP classification, which is based on suture diameter, knot security, and the nature of the suture material structure, among other properties, threads manufactured with different concentrations of sorbitol and cellulose can be considered suture material, suture USP 4 (diameter between 0.7 and 0.79 mm) [45,46,47]. 

Based on the results, the thread prototypes require the presence of sorbitol and cellulose. Nevertheless, after a given value of cellulose concentration, it should not be increased since the flexibility, hardness, and solubility become inadequate. Therefore, the cellulose in the threads should be less than 0.58 wt%. Another three prototypes were manufactured (Figure 11) with CS, 0.58 wt% of sorbitol, and three concentrations of cellulose (0.19 wt%, 0.29 wt%, and 0.39 wt%). By modifying the cellulose concentration, the characteristics of the threads changed. With a higher cellulose concentration, excellent consistency, texture, malleability, and tensile strength were obtained (1). Decreasing the cellulose concentration by half, the thread presented greater flexibility with good elongation capacity (2). When having a concentration greater than 50% of the initial, a thread with greater rigidity, low malleability, and little resistance to the tension was obtained. 

The thread prototypes that presented the best physical properties and good mechanical characteristics were obtained with 5.03 wt% corn starch, 19.34 wt% mucilage, 0.58 wt% sorbitol, 0.97 wt% cellulose, and 9.67 wt% glycerol in 64.41 wt% in water. Moreover, the thread formed a monofilament suture; having multifilament saturation in the thread is not desired due to the passage of microorganisms being favored [3,42]. Moreover, a monofilament suture benefits the healing process of wounds by having less rigidity and fragility, avoiding loss of tension, and being an absorbable biological thread; these are necessary features in medical applications [3,39,42,43]. The selected surgical thread prototype complies with most of the characteristics mentioned, except for good tensile strength or toughness. To avoid this, sorbitol and cellulose were employed as a plasticizer in the surgical thread synthesis. 

Currently, natural cellulose fibers are mainly used for internal gastric sutures. They are characterized by gaining tensile strength when wet, losing between 50% of their tensile strength at six months, and retaining between 30 wt% and 40 wt% at two years [5,42]. Moreover, biological characteristics such as bacterial adherence and tissue reaction, or histocompatibility, were considered for the selection of biopolymers and type of thread [3,34,35]; considering that the bioplastic obtained for the elaboration of the thread is of organic origin (NM), it is expected that it will not have an adverse reaction in the patient [3,34,35].

The physicochemical, functional, and rheological differences between cereal and fruit starches (in this research cereal starch was employed) are due to their different size, shape, and crystalline arrangement, the reason for which they can have diverse applications. Cereal starch presents a better alternative to making bioplastics or solid materials, while fruit starch works better for making gels or emulsifiers. Regarding the prototype thread and the possibility for biomedical use (suture thread), there is still a need to improve the tensile strength or use some class of coating (wax or paraffin) to prevent the tendency to desquamation (release of suture particles in the wound), or adding chitin, since it contributes with advantages for healing wounds (accelerated healing).

## 4. Conclusions

In this study, the extraction of NM was carried out using ethyl alcohol as a solvent; the experimental conditions allowed us to obtain a 50% higher yield than the yields reported in the literature. The bioplastics obtained were based on NM, CS, and glycerol as the plasticizer. The variation in concentration of NM and CS resulted in eight prototypes. From these, B1 and B2 have the appropriate characteristics for consideration in surgical thread manufacture. The resistance to solvents and solubility with water of these biopolymeric films (B1 and B2) were also analyzed, showing a lower degradation and solubility for B2 when compared to B1. Therefore, the concentrations of the base reagents (starch, glycerol, and mucilage) were similar to B2, and the incorporation of cellulose and sorbitol was studied for surgical thread synthesis. The thread prototype that presented the best physical properties (tensile strength and flexibility for good handling) was obtained using 5.03 wt% corn starch, 19.34 wt% mucilage, 0.58 wt% sorbitol, 0.97 wt% cellulose, and 9.67 wt% glycerol in 64.41 wt% distilled water. Compared to other prototypes reported in the literature, a surgical thread with uniform diameter, good elasticity, and low capillarity was obtained. In addition, the fact that its components are inert materials reduces the probability of an allergic reaction by the patient. These results indicate that it is possible to perform in vitro and in vivo tests with the obtained thread to evaluate its biocompatibility and viability in medical applications.

## Figures and Tables

**Figure 1 polymers-15-02112-f001:**
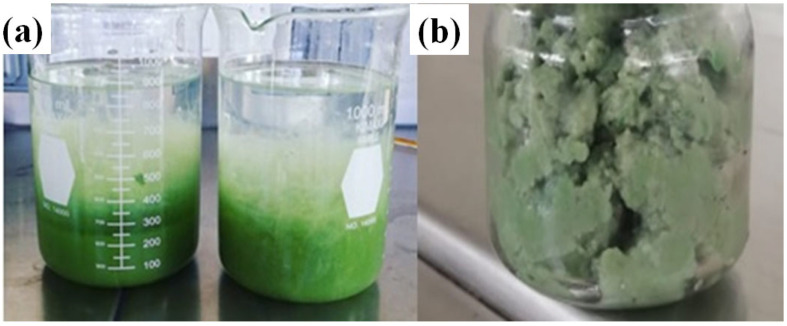
Images of the mucilage extraction: (**a**) ethanol blended with NM and (**b**) sediment.

**Figure 2 polymers-15-02112-f002:**
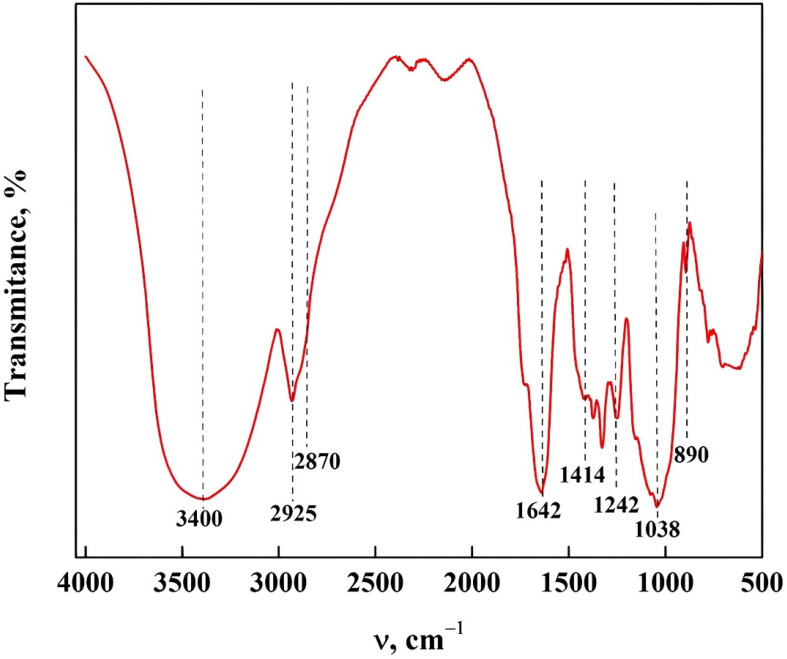
Characterization of NM by ATR-FTIR.

**Figure 3 polymers-15-02112-f003:**
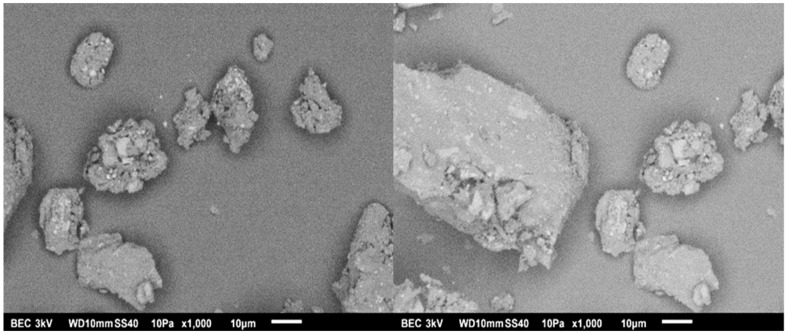
SEM images of NM.

**Figure 4 polymers-15-02112-f004:**
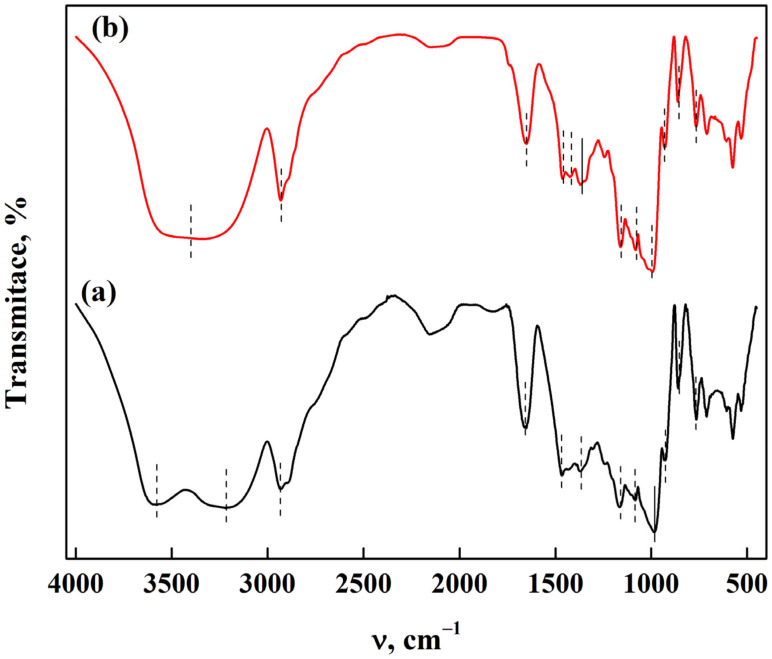
ATR-FTIR characterization of starch from (**a**) potato and (**b**) corn.

**Figure 5 polymers-15-02112-f005:**
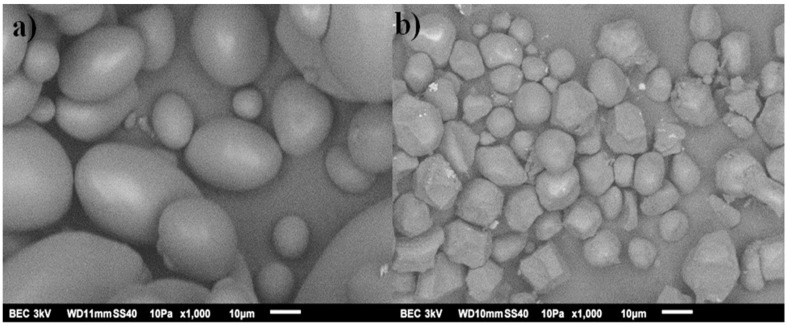
SEM images of starch from (**a**) potato and (**b**) corn.

**Figure 6 polymers-15-02112-f006:**
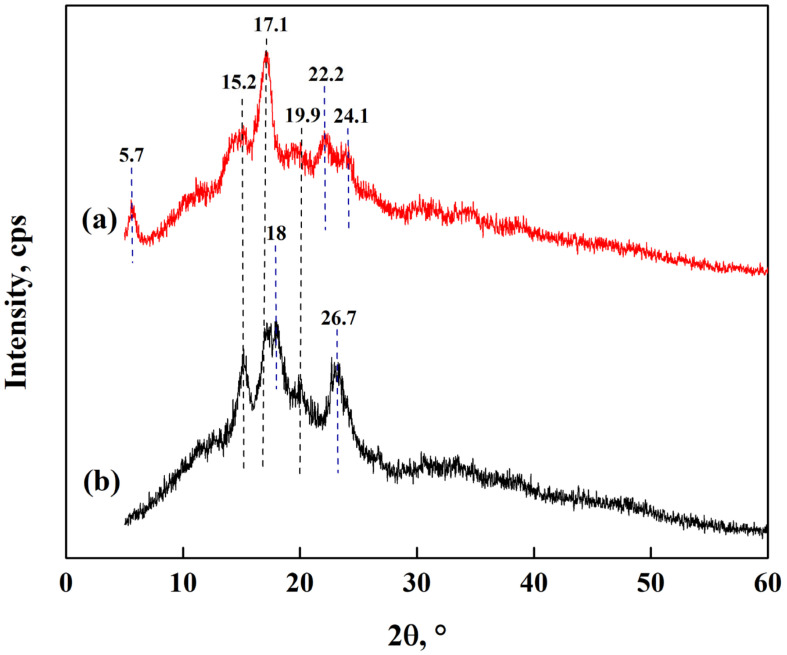
XRD patterns of starch from (**a**) potato and (**b**) corn.

**Figure 7 polymers-15-02112-f007:**
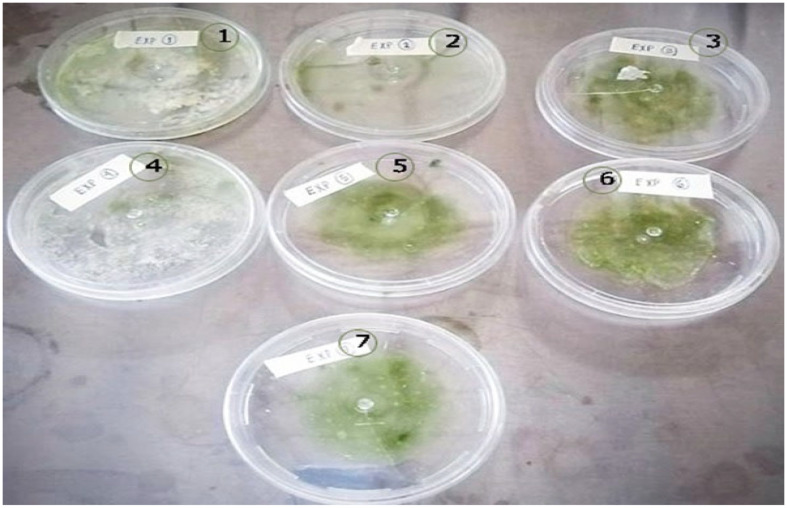
Images of the BP prepared with NM, CS, and glycerol at different proportions and dried at 40 °C for 24 h. The label number corresponds to the experiment number in Table 1.

**Figure 8 polymers-15-02112-f008:**
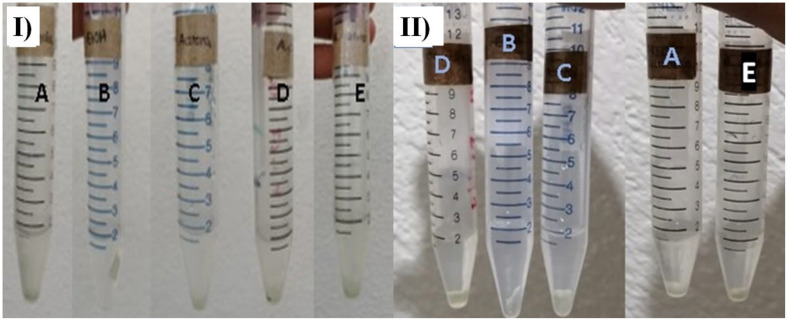
Tests conducted to explore the solubility of the biofilm, (**I**) E3 and (**II**) E5 after 12 h of contact with different solvents at 25 °C. Solvents: (**A**) distilled water, (**B**) ethyl alcohol, (**C**) acetone, (**D**) hydrochloric acid, and (**E**) acetic acid.

**Figure 9 polymers-15-02112-f009:**
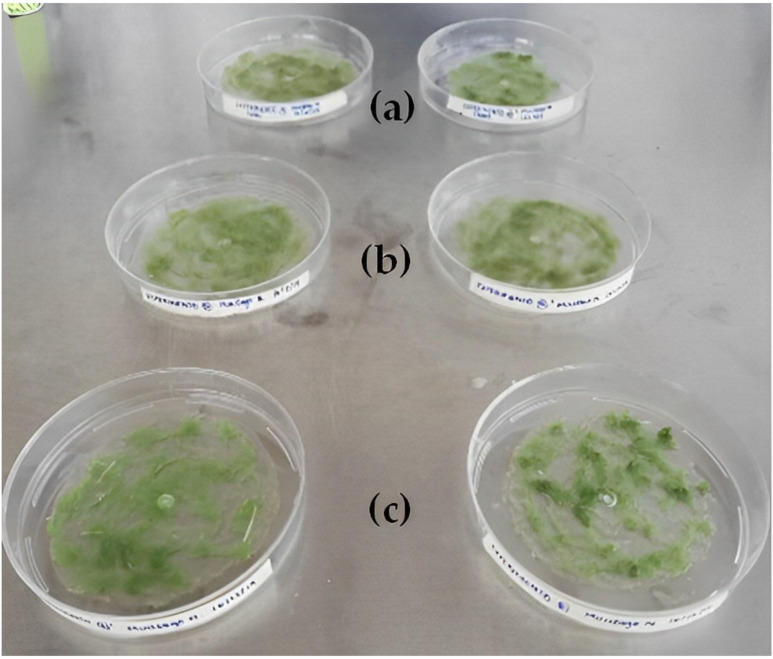
Bioplastic production using different starch sources (**a**) SAPS, (**b**) CS, and (**c**) BAPS.

**Figure 10 polymers-15-02112-f010:**
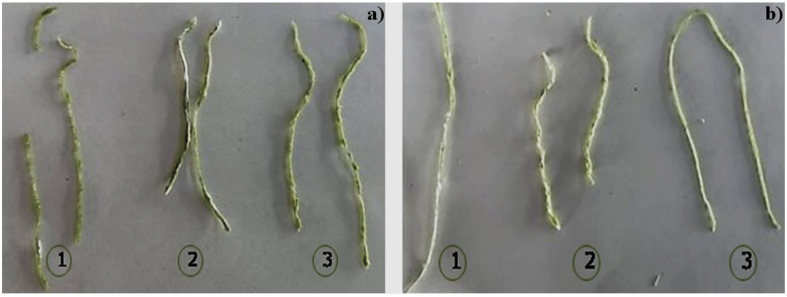
Manufacture threads with CS and cellulose: (**a**) with sorbitol and (**b**) without sorbitol. (1) Cellulose 0.58 wt%, (2) Cellulose 0.97 wt% and (3) Cellulose 1.93 wt%.

**Figure 11 polymers-15-02112-f011:**
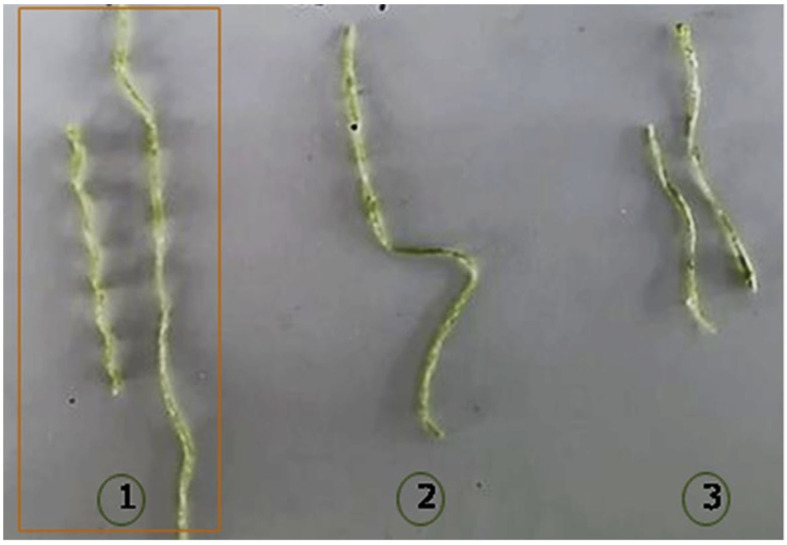
Manufacture of thread with sorbitol, cellulose, and CS. (1) Cellulose 0.19 wt%, (2) Cellulose 0.29 wt% and (3) Cellulose 0.39 wt%. The best thread is marked in the square.

**Table 1 polymers-15-02112-t001:** Mixtures employed to generate the BP.

Experiment	Mucilage (mg)	H_2_O (mL)
1	1.05	10
2	1.03	20
3	1.04	30
4	1.01	5
5	1.02	15
6	1.53	30
7	1.51	10

**Table 2 polymers-15-02112-t002:** Components and proportions employed during the preparation of surgical thread.

Starch	Sorbitol (mg)	Cellulose (mg)
Corn (CS)	10	10
		50
		100
	20	10
		20
		50
		100
	30	10
		20
		50
		100
Potato of Sigma Aldrich (SAPS)	10	10
		50
		100
	20	10
		20
		50
		100
	30	10
		20
		50
		100
Potato of Baker Analyzed (BAPS)	10	10
		50
		100
	20	10
		20
		50
		100
	30	10
		20
		50
		100

**Table 3 polymers-15-02112-t003:** Comparison of the bioplastic solubility with different solvents.

Solvent	Solubility	
	B1	B2
(**A**) Distilled water	Completely soluble	Completely soluble
(**B**) Ethyl alcohol	Insoluble	Insoluble
(**C**) Acetone	Slightly soluble	Slightly soluble
(**D**) Hydrochloric acid (0.1 M)	Completely soluble	Slightly soluble
(**E**) Acetic acid (0.1 M)	Highly soluble	Highly soluble

**Table 4 polymers-15-02112-t004:** Experimental results of water solubility of the manufactured bioplastics.

Bioplastic	Initial Weight (g)	Initial Dry Weight (W_ds_; g)	Final Dry Weight (W_df_; g)	Solubility (%)
B1	0.367	0.347	0.110	68.30
B2	0.253	0.170	0.118	30.59

**Table 5 polymers-15-02112-t005:** Mechanical properties of bioplastics.

Bioplastic	σ_F_, MPa	E_F_, MPa	σ_T_, MPa	E_T_, MPa
B1	2.71	278.4	1.87	340.1
B2	4.12	333.2	3.97	373.2

## Data Availability

The data presented in this study are available on request from the corresponding author.

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
