# Peer review of "Preparation of Surgical Thread from a Bioplastic Based on Nopal Mucilage"

_polymers, 2023, doi:10.3390/polym15092112_

Round 1
Reviewer 1 Report (Previous Reviewer 3)
It should be accepted in its present form
Author Response
Dear Reviewer, thank you for taking the time to read and review our manuscript.
Kind regards
Reviewer 2 Report (New Reviewer)
1. What natural material was used in the study to obtain bioplastics?
2. How was the mucilage extracted from nopal (NM)?
3. What properties were determined during the characterization of the extract?
4. What differences were observed in the structure of the polymeric chains of corn starch (CS) and potato starch (PS)?
5. How was the degradation of the biopolymers with the use of different solvents performed?
6. What were the proportions of sorbitol and cellulose added to the BP mixtures to synthesize the surgical thread?
7. What were the characteristics of the obtained surgical thread?
8. How did the obtained surgical thread compare to other prototypes reported in the literature?
9. What specific medical applications is this thread intended for?
10. Can you provide more details on the changes in physical properties that were observed when modifying the proportion of components in the mixture?
11. What are the potential advantages and disadvantages of using sorbitol and cellulose as plasticizers in the thread?
12. Have these threads been tested for biocompatibility and safety in vivo?
13. How does this thread compare to other materials currently used in medical applications?
14. Are there plans to scale up the production of these threads for commercial use?
15. What were the degradation properties of the thread in different solvents, and how might this affect its longevity and effectiveness in medical applications?
16. Can you describe the manufacturing process for these threads and how easily they can be produced on a large scale?
17. Has the thread been tested for any specific medical applications yet, and if so, what were the results?
18. What are the potential cost implications of using this thread in medical applications compared to other materials?
19. What are the different variables that can affect the yield of NM mentioned in the study?
20. How was the yield of NM increased in this study compared to reports in the literature?
21. What modifications were made to increase the yield of NM in this study?
22. Why was it not required to dry the precipitate obtained in this study?
23. How do the yields reported in this study compare to those reported in the literature for the use of solvents for the precipitation of NM?
24. What was the apparent viscosity obtained for the NM in this study and at what pH value?
25. What is the relationship between the pH value and the viscosity of the extract according to the literature?
26. What type of fluid is considered a non-Newtonian fluid with characteristics of a pseudoplastic fluid according to the literature?
27. What are the implications of the increased yield of NM reported in this study for potential industrial applications?
28. Are there any potential drawbacks or limitations to the modifications made in this study to increase the yield of NM?
Author Response
Dear reviewer, thank you for your comments; we hope your answers have adequately answered your questions.

This manuscript is a resubmission of an earlier submission. The following is a list of the peer review reports and author responses from that submission.
Round 1
Reviewer 1 Report
The manuscript entitiled "Synthesis of bioplastic from nopal mucilage for the manufacture of surgical thread" presents the study about prepration of biopolymer badsed materials with potential application as surgical thread. After careful review of the manuscript, I regrate to say that the manuscript neither presents significant novelty nor the scientifically sound approach to meet the necessary rigor to warrant the publication in high impact factor journal like Polymers.
Author Response
Dear reviewer thanks you for your comments
Reviewer 2 Report
1. Abstract appears to be fine, except the last sentence (Line no 27-29). Authors are recommended to reframe the sentence. The concluding remark of abstract should highlight the essence of the work conducted. Considering the significance of work conducted, the conclusion of abstract can be surely made more effective.
2. material and methodology
initial lines of 2.5 are complex, see text highlighted in yellow. Can be divided into two sentences for better clarity.
3. results and discussion
Adequate experimental studies have been mentioned in the study. However, the manuscript at places indicate casual approach towards scientific writing. some sentences have been highlighted in manuscript which are to be rewritten for effective impact. Authors are advised to throughly read the manuscript for improving the language.
Also in results and discussion at places discussion appears to overshadow the result. Discussion is ellaborate ( which is good), compared to that at specific segments results of the work could have been explained in more detail which would further enhance the significance of the study. For example when citing the tables and figures.
2. Authors to throughly check all references cited, Some references are not in format. refer to highlighted portion in manuscript. Ref. 30 year of publication is mentioned twice. Pls check whether ref. 34 and 35 are same or different. in ref 34 only author names are mentioned and in ref 35 all other detail except author names if given. If same then cited ref no in manuscript to be changed accordingly.

Author Response
Dear reviewer than for your comments and suggestions towards the manuscript, all of them were taken into account to improve the work
Reviewer 3 Report
Please find the comments and suggestions in attachment

Author Response

(The authors gave the same response as above.)

Reviewer 4 Report
Dear Authors
Your manuscript is well-organized and written.
The topic is novel and original and there is not a lot such articles on this topic.
However, please review newer references in your Introduction part related to materials used in your study.
Experimental part is well described. The Results section consequently follows the experiments and finally Conclusion is well done.
The attached file contains minor revision of the manuscript. Please take them in consideration in order to improve your article.
Thanks.

Author Response

(The authors gave the same response as above.)

Round 2
Reviewer 1 Report
I appreciate authors' attempt to improve the manuscript in the revised version. However, I regret to mention that these purfunctory revisions in different sections of the manuscript do not address more fundamental flaws of the manuscript in terms of novelty and scientific approach.
Issues with novelty:
Although the authors claim that the materials ("surgical sutures) fabricated are based on Nopal Mucilage (NP), what I can see from the data presented in the table Table 1, NP appears to be an additive (ranging between 1 and 1.5 mg) in the matrix of Starch. Fabrication of starch and cellulose based based materials (mebranes, films and thereads) with different potential application if well studied and available in the literature. Interestingly, the authors present ratio of volum% of starch in the water, how they calculated the volume/volume% of starch in the formulations presentated on the Table 1. It woud have been much more easier for a reader if the startch amount were presented in mass.
The authors haven't compared what NM is doining in the in the formulations by comparision. It would have been nicer to see the material without NM for comparision showing improtance of NM.
Issues with scientific approach:
Authors do not provide enough information or worse in some cases provide no relevant information at all about the materials they have used. For example, glycerol, sorbitol and cellulsoe. What kind of cellulose? Plant derived cellulose?, bacterial cellulose, or cellulose derivates such as carboxymethyl cellulose? The information is relevant because authors present in the table that the fabricated material is completley soluble in water for E3 and E5 (although it is not clear that hwo E3 was different from E5 and how E3 and E5 were different from reset of others). How a material that contains cellulose is completely soluble in water unless it is cellulose derivative?